# Variational Information Bottleneck for Unsupervised Clustering: Deep Gaussian Mixture Embedding

**DOI:** 10.3390/e22020213

**Published:** 2020-02-13

**Authors:** Yiğit Uğur, George Arvanitakis, Abdellatif Zaidi

**Affiliations:** 1Laboratoire d’informatique Gaspard-Monge, Université Paris-Est, 77454 Champs-sur-Marne, France; 2Mathematical and Algorithmic Sciences Lab, Paris Research Center, Huawei Technologies, 92100 Boulogne-Billancourt, France

**Keywords:** clustering, unsupervised learning, Gaussian mixture model, information bottleneck

## Abstract

In this paper, we develop an unsupervised generative clustering framework that combines the variational information bottleneck and the Gaussian mixture model. Specifically, in our approach, we use the variational information bottleneck method and model the latent space as a mixture of Gaussians. We derive a bound on the cost function of our model that generalizes the Evidence Lower Bound (ELBO) and provide a variational inference type algorithm that allows computing it. In the algorithm, the coders’ mappings are parametrized using neural networks, and the bound is approximated by Markov sampling and optimized with stochastic gradient descent. Numerical results on real datasets are provided to support the efficiency of our method.

## 1. Introduction

Clustering consists of partitioning a given dataset into various groups (clusters) based on some similarity metric, such as the Euclidean distance, $L_{1}$ norm, $L_{2}$ norm, $L_{\infty }$ norm, the popular logarithmic loss measure, or others. The principle is that each cluster should contain elements of the data that are closer to each other than to any other element outside that cluster, in the sense of the defined similarity measure. If the joint distribution of the clusters and data is not known, one should operate blindly in doing so, i.e., using only the data elements at hand; and the approach is called unsupervised clustering [1,2]. Unsupervised clustering is perhaps one of the most important tasks of unsupervised machine learning algorithms currently, due to a variety of application needs and connections with other problems.

Clustering can be formulated as follows. Consider a dataset that is composed of *N* samples ${\left\{x_{i}\right\}}_{i=1}^{N}$, which we wish to partition into |C|≥1 clusters. Let C={1,⋯,|C|} be the set of all possible clusters and *C* designate a categorical random variable that lies in C and stands for the index of the actual cluster. If X is a random variable that models elements of the dataset, given that $X=x_{i}$ induces a probability distribution on *C*, which the learner should learn, thus mathematically, the problem is that of estimating the values of the unknown conditional probability $P_{C|X}\left(\cdot |x_{i}\right)$ for all elements $x_{i}$ of the dataset. The estimates are sometimes referred to as the assignment probabilities.

Examples of unsupervised clustering algorithms include the very popular *K*-means [3] and Expectation Maximization (EM) [4]. The *K*-means algorithm partitions the data in a manner that the Euclidean distance among the members of each cluster is minimized. With the EM algorithm, the underlying assumption is that the data comprise a mixture of Gaussian samples, namely a Gaussian Mixture Model (GMM); and one estimates the parameters of each component of the GMM while simultaneously associating each data sample with one of those components. Although they offer some advantages in the context of clustering, these algorithms suffer from some strong limitations. For example, it is well known that the *K*-means is highly sensitive to both the order of the data and scaling; and the obtained accuracy depends strongly on the initial seeds (in addition to that, it does not predict the number of clusters or *K*-value). The EM algorithm suffers mainly from slow convergence, especially for high-dimensional data.

Recently, a new approach has emerged that seeks to perform inference on a transformed domain (generally referred to as latent space), not the data itself. The rationale is that because the latent space often has fewer dimensions, it is more convenient computationally to perform inference (clustering) on it rather than on the high-dimensional data directly. A key aspect then is how to design a latent space that is amenable to accurate low-complexity unsupervised clustering, i.e., one that preserves only those features of the observed high-dimensional data that are useful for clustering while removing all redundant or non-relevant information. Along this line of work, we can mention [5], which utilized Principal Component Analysis (PCA) [6,7] for dimensionality reduction followed by *K*-means for clustering the obtained reduced dimension data; or [8], which used a combination of PCA and the EM algorithm. Other works that used alternatives for the linear PCA include kernel PCA [9], which employs PCA in a non-linear fashion to maximize variance in the data.

Thisby’s Information Bottleneck (IB) method [10] formulates the problem of finding a good representation U that strikes the right balance between capturing all information about the categorical variable *C* that is contained in the observation X and using the most concise representation for it. The IB problem can be written as the following Lagrangian optimization:(1)$$\min_{P_{U|X}}\;I\left(X;U\right)-sI\left(C;U\right)\;,$$
where I(·;·) denotes Shannon’s mutual information and *s* is a Lagrange-type parameter, which controls the trade-off between accuracy and regularization. In [11,12], a text clustering algorithm is introduced for the case in which the joint probability distribution of the input data is known. This text clustering algorithm uses the IB method with an annealing procedure, where the parameter *s* is increased gradually. When s→0, the representation U is designed with the most compact form, i.e., |U|=1, which corresponds to the maximum compression. By gradually increasing the parameter *s*, the emphasis on the relevance term I(C;U) increases, and at a critical value of *s*, the optimization focuses on not only the compression, but also the relevance term. To fulfill the demand on the relevance term, this results in the cardinality of U bifurcating. This is referred as a phase transition of the system. The further increases in the value of *s* will cause other phase transitions, hence additional splits of *U* until it reaches the desired level, e.g., |U|=|C|.

However, in the real-world applications of clustering with large-scale datasets, the joint probability distributions of the datasets are unknown. In practice, the usage of Deep Neural Networks (DNN) for unsupervised clustering of high-dimensional data on a lower dimensional latent space has attracted considerable attention, especially with the advent of Autoencoder (AE) learning and the development of powerful tools to train them using standard backpropagation techniques [13,14]. Advanced forms include Variational Autoencoders (VAE) [13,14], which are generative variants of AE that regularize the structure of the latent space, and the more general Variational Information Bottleneck (VIB) of [15], which is a technique that is based on the information bottleneck method and seeks a better trade-off between accuracy and regularization than VAE via the introduction of a Lagrange-type parameter *s*, which controls that trade-off and whose optimization is similar to deterministic annealing [12] or stochastic relaxation.

In this paper, we develop an unsupervised generative clustering framework that combines VIB and the Gaussian mixture model. Specifically, in our approach, we use the variational information bottleneck method and model the latent space as a mixture of Gaussians. We derive a bound on the cost function of our model that generalizes the Evidence Lower Bound (ELBO) and provide a variational inference type algorithm that allows computing it. In the algorithm, the coders’ mappings are parametrized using Neural Networks (NN), and the bound is approximated by Markov sampling and optimized with stochastic gradient descent. Furthermore, we show how tuning the hyper-parameter *s* appropriately by gradually increasing its value with iterations (number of epochs) results in a better accuracy. Furthermore, the application of our algorithm to the unsupervised clustering of various datasets, including the MNIST [16], REUTERS [17], and STL-10 [18], allows a better clustering accuracy than previous state-of-the-art algorithms. For instance, we show that our algorithm performs better than the Variational Deep Embedding (VaDE) algorithm of [19], which is based on VAE and performs clustering by maximizing the ELBO. Our algorithm can be seen as a generalization of the VaDE, whose ELBO can be recovered by setting s=1 in our cost function. In addition, our algorithm also generalizes the VIB of [15], which models the latent space as an isotropic Gaussian, which is generally not expressive enough for the purpose of unsupervised clustering. Other related works, which are of lesser relevance to the contribution of this paper, are the Deep Embedded Clustering (DEC) of [20] and the Improved Deep Embedded Clustering (IDEC) of [21,22]. For a detailed survey of clustering with deep learning, the readers may refer to [23].

To the best of our knowledge, our algorithm performs the best in terms of clustering accuracy by using deep neural networks without any prior knowledge regarding the labels (except the usual assumption that the number of classes is known) compared to the state-of-the-art algorithms of the unsupervised learning category. In order to achieve the outperforming accuracy: (i) we derive a cost function that contains the IB hyperparameter *s* that controls optimal trade-offs between the accuracy and regularization of the model; (ii) we use a lower bound approximation for the KL term in the cost function, that does not depend on the clustering assignment probability (note that the clustering assignment is usually not accurate in the beginning of the training process); and (iii) we tune the hyperparameter *s* by following an annealing approach that improves both the convergence and the accuracy of the proposed algorithm.

Throughout this paper, we use the following notation. Uppercase letters are used to denote random variables, e.g., *X*; lowercase letters are used to denote realizations of random variables, e.g., *x*; and calligraphic letters denote sets, e.g., X. The cardinality of a set X is denoted by |X|. Probability mass functions (pmfs) are denoted by $P_{X}\left(x\right)=\mathrm{Pr}\left\{X=x\right\}$ and, sometimes, for short, as p(x). Boldface uppercase letters denote vectors or matrices, e.g., X, where context should make the distinction clear. We denote the covariance of a zero mean, complex-valued, vector X by $\Sigma_{x}=E\left[{\mathrm{XX}}^{†}\right]$, where ${\left(\cdot \right)}^{†}$ indicates the conjugate transpose. For random variables *X* and *Y*, the entropy is denoted as H(X), i.e., $H\left(X\right)=E_{P_{X}}\left[-\log P_{X}\right]$, and the mutual information is denoted as I(X;Y), i.e., $I\left(X;Y\right)=H\left(X\right)-H\left(X|Y\right)=H\left(Y\right)-H\left(Y|X\right)=E_{P_{X,Y}}\left[\log\frac{P_{X,Y}}{P_{X}P_{Y}}\right]$. Finally, for two probability measures $P_{X}$ and $Q_{X}$ on a random variable X∈X, the relative entropy or Kullback–Leibler divergence is denoted as $D_{\mathrm{KL}}\left(P_{X}∥Q_{X}\right)$, i.e., $D_{\mathrm{KL}}\left(P_{X}∥Q_{X}\right)=E_{P_{X}}\left[\log\frac{P_{X}}{Q_{X}}\right]$.

## 2. Proposed Model

In this section, we explain the proposed model, the so-called Variational Information Bottleneck with Gaussian Mixture Model (VIB-GMM), in which we use the VIB framework and model the latent space as a GMM. The proposed model is depicted in Figure 1, where the parameters $\pi_{c}$, $\mu_{c}$, $\Sigma_{c}$, for all values of c∈C, are to be optimized jointly with those of the employed NNs as instantiation of the coders. Furthermore, the assignment probabilities are estimated based on the values of latent space vectors instead of the observations themselves, i.e., $P_{C|X}=Q_{C|U}$. In the rest of this section, we elaborate on the inference and generative network models for our method.

### 2.1. Inference Network Model

We assume that observed data x are generated from a GMM with |C| components. Then, the latent representation u is inferred according to the following procedure:One of the components of the GMM is chosen according to a categorical variable *C*.  The data x are generated from the *c*th component of the GMM, i.e., $P_{X|C}∼N\left(x;{\tilde{\mu }}_{c},{\tilde{\Sigma }}_{c}\right)$.  Encoder maps x to a latent representation u according to $P_{U|X}∼N\left(\mu_{\theta },\Sigma_{\theta }\right)$.
3.1.The encoder is modeled with a DNN $f_{\theta }$, which maps x to the parameters of a Gaussian distribution, i.e., $\left[\mu_{\theta },\Sigma_{\theta }\right]=f_{\theta }\left(x\right)$.  3.2.The representation u is sampled from $N(\mu_{\theta },\Sigma_{\theta })$.

For the inference network, shown in Figure 2, the following Markov chain holds: 
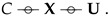
(2)

### 2.2. Generative Network Model

Since the encoder extracts useful representations of the dataset and we assume that the dataset is generated from a GMM, we model our latent space also with a mixture of Gaussians. To do so, the categorical variable *C* is embedded with the latent variable U. The reconstruction of the dataset is generated according to the following procedure:One of the components of the GMM is chosen according to a categorical variable *C*, with a prior distribution $Q_{C}$.  The representation u is generated from the *c*th component, i.e., $Q_{U|C}∼N\left(u;\mu_{c},\Sigma_{c}\right)$.  The decoder maps the latent representation u to $\hat{x}$, which is the reconstruction of the source x by using the mapping $Q_{X|U}$.
3.1The decoder is modeled with a DNN $g_{\phi }$ that maps u to the estimate $\hat{x}$, i.e., $\left[\hat{x}\right]=g_{\phi }\left(u\right)$.

For the generative network, shown in Figure 3, the following Markov chain holds: 
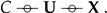
(3)

## 3. Proposed Method

In this section, we present our clustering method. First, we provide a general cost function for the problem of the unsupervised clustering that we study here based on the variational IB framework; and we show that it generalizes the ELBO bound developed in [19]. We then parametrize our model using NNs whose parameters are optimized jointly with those of the GMM. Furthermore, we discuss the influence of the hyperparameter *s* that controls optimal trade-offs between accuracy and regularization.

### 3.1. Brief Review of Variational Information Bottleneck for Unsupervised Learning

As described in Section 2, the stochastic encoder $P_{U|X}$ maps the observed data x to a representation u. Similarly, the stochastic decoder $Q_{X|U}$ assigns an estimate $\hat{x}$ of x based on the vector u. As per the IB method [10], a suitable representation U should strike the right balance between capturing all information about the categorical variable *C* that is contained in the observation X and using the most concise representation for it. This leads to maximizing the following Lagrange problem:(4)$$\;L_{s}\left(P\right)=I\left(C;U\right)-sI\left(X;U\right)\;,$$
where s≥0 designates the Lagrange multiplier and, for convenience, P denotes the conditional distribution $P_{U|X}$.

Instead of (Equation 4), which is not always computable in our unsupervised clustering setting, we find it convenient to maximize an upper bound of $L_{s}\left(P\right)$ given by:(5)$$\;{\tilde{L}}_{s}\left(P\right):=I\left(X;U\right)-sI\left(X;U\right)\;\overset{\left(a\right)}{=}H\left(X\right)-H\left(X|U\right)-s\left[H\left(U\right)-H\left(U|X\right)\right]\;,$$
where (a) is due to the definition of mutual information (using the Markov chain 
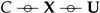
, it is easy to see that ${\tilde{L}}_{s}\left(P\right)\geq L_{s}\left(P\right)$ for all values of P). Noting that H(X) is constant with respect to $P_{U|X}$, maximizing ${\tilde{L}}_{s}\left(P\right)$ over P is equivalent to maximizing: (6)$$\begin{array}{cc} L_{s}^{\prime }\left(P\right): & =-H(X|U)-s[H(U)-H(U|X)] \end{array}$$(7)$$\begin{array}{cc} \; & =E_{P_{X}}\left[E_{P_{U|X}}\left[\log P_{X|U}+s\log P_{U}-s\log P_{U|X}\right]\right]\;. \end{array}$$
For a variational distribution $Q_{U}$ on U (instead of the unknown $P_{U}$) and a variational stochastic decoder $Q_{X|U}$ (instead of the unknown optimal decoder $P_{X|U}$), let $Q:=\{Q_{X|U},Q_{U}\}$. Furthermore, let:(8)$$\;L_{s}^{\mathrm{VB}}\left(P,Q\right):=E_{P_{X}}\left[E_{P_{U|X}}\left[\log Q_{X|U}\right]-sD_{\mathrm{KL}}\left(P_{U|X}∥Q_{U}\right)\right]\;.$$

**Lemma** **1.**
*For given P, we have:*
$$L_{s}^{\mathrm{VB}}\left(P,Q\right)\leq L_{s}^{\prime }\left(P\right)\;,\;for\;all\;Q\;.$$
*In addition, there exists a unique Q that achieves the maximum $\max_{Q}L_{s}^{\mathrm{VB}}\left(P,Q\right)=L_{s}^{\prime }\left(P\right)$ and is given by:*
$$Q_{X|U}^{*}=P_{X|U}\;,\;Q_{U}^{*}=P_{U}\;.$$


**Proof.** The proof of Lemma 1 is given in Appendix A.

Using Lemma 1, maximization of (Equation 6) can be written in term of the variational IB cost as follows:(9)$$\begin{array}{c} \;\max_{P}L_{s}^{\prime }\left(P\right)=\max_{P}\max_{Q}L_{s}^{\mathrm{VB}}\left(P,Q\right)\;. \end{array}$$
Next, we develop an algorithm that solves the maximization problem (Equation 9), where the encoding mapping $P_{U|X}$, the decoding mapping $Q_{X|U}$, as well as the prior distribution of the latent space $Q_{U}$ are optimized jointly.

**Remark** **1.**
*As we already mentioned in the beginning of this section, the related work [19] performed unsupervised clustering by combining VAE with GMM. Specifically, it maximizes the following ELBO bound:*
(10)$$\;L_{1}^{\mathrm{VaDE}}:=E_{P_{X}}\left[E_{P_{U|X}}\left[\log Q_{X|U}\right]-D_{\mathrm{KL}}\left(P_{C|X}∥Q_{C}\right)-E_{P_{C|X}}\left[D_{\mathrm{KL}}\left(P_{U|X}∥Q_{U|C}\right)\right]\right].$$
*Let, for an arbitrary non-negative parameter s, $L_{s}^{\mathrm{VaDE}}$ be a generalization of the ELBO bound in *(Equation 10)* of [19] given by:*
(11)$$\;L_{s}^{\mathrm{VaDE}}:=E_{P_{X}}\left[E_{P_{U|X}}\left[\log Q_{X|U}\right]-sD_{\mathrm{KL}}\left(P_{C|X}∥Q_{C}\right)-sE_{P_{C|X}}\left[D_{\mathrm{KL}}\left(P_{U|X}∥Q_{U|C}\right)\right]\right].$$

*Investigating the right-hand side (RHS) of *(Equation 11)*, we get:*
(12)$$\;L_{s}^{\mathrm{VB}}\left(P,Q\right)=L_{s}^{\mathrm{VaDE}}+sE_{P_{X}}\left[E_{P_{U|X}}\left[D_{\mathrm{KL}}\left(P_{C|X}∥Q_{C|U}\right)\right]\right]\;,$$
*where the equality holds since:*
(13)$$\begin{array}{cc} L_{s}^{\mathrm{VaDE}} & =E_{P_{X}}\left[E_{P_{U|X}}\left[\log Q_{X|U}\right]-sD_{\mathrm{KL}}\left(P_{C|X}∥Q_{C}\right)-sE_{P_{C|X}}\left[D_{\mathrm{KL}}\left(P_{U|X}∥Q_{U|C}\right)\right]\right] \end{array}$$
(14)$$\begin{array}{cc} \; & \overset{\left(a\right)}{=}E_{P_{X}}[E_{P_{U|X}}\left[\log Q_{X|U}\right]-sD_{\mathrm{KL}}\left(P_{U|X}∥Q_{U}\right)-sE_{P_{U|X}}\left[D_{\mathrm{KL}}\left(P_{C|X}∥Q_{C|U}\right)\right] \end{array}$$
(15)$$\begin{array}{cc} \; & \overset{\left(b\right)}{=}L_{s}^{\mathrm{VB}}\left(P,Q\right)-sE_{P_{X}}\left[E_{P_{U|X}}\left[D_{\mathrm{KL}}\left(P_{C|X}∥Q_{C|U}\right)\right]\right]\;, \end{array}$$
*where (a) can be obtained by expanding and re-arranging terms under the Markov chain 
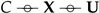
 (for a detailed treatment, please look at Appendix B); and (b) follows from the definition of $L_{s}^{\mathrm{VB}}\left(P,Q\right)$ in *(Equation 8)*.*

*Thus, by the non-negativity of relative entropy, it is clear that $L_{s}^{\mathrm{VaDE}}$ is a lower bound on $L_{s}^{\mathrm{VB}}\left(P,Q\right)$. Furthermore, if the variational distribution Q is such that the conditional marginal $Q_{C|U}$ is equal to $P_{C|X}$, the bound is tight since the relative entropy term is zero in this case.*


### 3.2. Proposed Algorithm: VIB-GMM

In order to compute (Equation 9), we parametrize the distributions $P_{U|X}$ and $Q_{X|U}$ using DNNs. For instance, let the stochastic encoder $P_{U|X}$ be a DNN $f_{\theta }$ and the stochastic decoder $Q_{X|U}$ be a DNN $g_{\phi }$. That is:(16)$$\;\begin{array}{cc} \; & P_{\theta }\left(u|x\right)=N\left(u;\mu_{\theta },\Sigma_{\theta }\right)\;,\;\mathrm{where}\;\left[\mu_{\theta },\Sigma_{\theta }\right]=f_{\theta }\left(x\right)\;,\\ \; & Q_{\phi }\left(x|u\right)=g_{\phi }\left(u\right)=\left[\hat{x}\right]\;, \end{array}$$
where θ and ϕ are the weight and bias parameters of the DNNs. Furthermore, the latent space is modeled as a GMM with |C| components with parameters $\psi :={\left\{\pi_{c},\mu_{c},\Sigma_{c}\right\}}_{c=1}^{\left|C\right|}$, i.e.,
(17)$$\;Q_{\psi }\left(u\right)=\sum_{c}\pi_{c}\;N\left(u;\mu_{c},\Sigma_{c}\right)\;.$$
Using the parametrizations above, the optimization of (Equation 9) can be rewritten as:(18)$$\;\max_{\theta ,\phi ,\psi }\;L_{s}^{\mathrm{NN}}\left(\theta ,\phi ,\psi \right)\;$$
where the cost function $L_{s}^{\mathrm{NN}}\left(\theta ,\phi ,\psi \right)$ is given by:(19)$$\;L_{s}^{\mathrm{NN}}\left(\theta ,\phi ,\psi \right):=E_{P_{X}}\left[E_{P_{\theta }\left(U|X\right)}\left[\log Q_{\phi }\left(X|U\right)\right]-sD_{\mathrm{KL}}\left(P_{\theta }\left(U|X\right)∥Q_{\psi }\left(U\right)\right)\right]\;.$$
Then, for given observations of *N* samples, i.e., ${\left\{x_{i}\right\}}_{i=1}^{N}$, (Equation 18) can be approximated in terms of an empirical cost as follows:(20)$$\;\max_{\theta ,\phi ,\psi }\;\frac{1}{N}\sum_{i=1}^{N}L_{s,i}^{\mathrm{emp}}\left(\theta ,\phi ,\psi \right)\;,$$
where $L_{s,i}^{\mathrm{emp}}\left(\theta ,\phi ,\psi \right)$ is the empirical cost for the *i*th observation $x_{i}$ and given by:(21)$$\;L_{s,i}^{\mathrm{emp}}\left(\theta ,\phi ,\psi \right)=E_{P_{\theta }\left(U_{i}|X_{i}\right)}\left[\log Q_{\phi }\left(X_{i}|U_{i}\right)\right]-sD_{\mathrm{KL}}\left(P_{\theta }\left(U_{i}|X_{i}\right)∥Q_{\psi }\left(U_{i}\right)\right)\;.$$
Furthermore, the first term of the RHS of (Equation 21) can be computed using Monte Carlo sampling and the reparametrization trick [13]. In particular, $P_{\theta }\left(u|x\right)$ can be sampled by first sampling a random variable Z with distribution $P_{Z}$, i.e., $P_{Z}=N\left(0,I\right)$, then transforming the samples using some function ${\tilde{f}}_{\theta }:X\times Z\to U$, i.e., $u={\tilde{f}}_{\theta }\left(x,z\right)$. Thus,
$$E_{P_{\theta }\left(U_{i}|X_{i}\right)}\left[\log Q_{\phi }\left(X_{i}|U_{i}\right)\right]=\frac{1}{M}\sum_{m=1}^{M}\log q\left(x_{i}|u_{i,m}\right),\;u_{i,m}=\mu_{\theta ,i}+\Sigma_{\theta ,i}^{\frac{1}{2}}\cdot \epsilon_{m},\;\epsilon_{m}∼N\left(0,I\right)\;,$$
where *M* is the number of samples for the Monte Carlo sampling step.

The second term of the RHS of (Equation 21) is the KL divergence between a single component multivariate Gaussian and a Gaussian mixture model with |C| components. An exact closed-form solution for the calculation of this term does not exist. However, a variational lower bound approximation [24] of it can be obtained as:(22)$$\;D_{\mathrm{KL}}\left(P_{\theta }\left(U_{i}|X_{i}\right)∥Q_{\psi }\left(U_{i}\right)\right)=-\log\sum_{c=1}^{\left|C\right|}\pi_{c}\exp\left(-D_{\mathrm{KL}}(N\left(\mu_{\theta ,i},\Sigma_{\theta ,i}\right)∥N\left(\mu_{c},\Sigma_{c}\right)\right)\;.$$
In particular, in the specific case in which the covariance matrices are diagonal, i.e., $\Sigma_{\theta ,i}:=\mathrm{diag}\left({\left\{\sigma_{\theta ,i,j}^{2}\right\}}_{j=1}^{n_{u}}\right)$ and $\Sigma_{c}:=\mathrm{diag}\left({\left\{\sigma_{c,j}^{2}\right\}}_{j=1}^{n_{u}}\right)$, with $n_{u}$ denoting the latent space dimension, (Equation 22) can be computed as follows:(23)$$\begin{array}{c} D_{\mathrm{KL}}\left(P_{\theta }\left(U_{i}|X_{i}\right)∥Q_{\psi }\left(U_{i}\right)\right)\\ \;=-\log\sum_{c=1}^{\left|C\right|}\pi_{c}\exp\left(-\frac{1}{2}\sum_{j=1}^{n_{u}}\left[\frac{{\left(\mu_{\theta ,i,j}-\mu_{c,j}\right)}^{2}}{\sigma_{c,j}^{2}}+\log\frac{\sigma_{c,j}^{2}}{\sigma_{\theta ,i,j}^{2}}-1+\frac{\sigma_{\theta ,i,j}^{2}}{\sigma_{c,j}^{2}}\right]\right)\;, \end{array}$$
where $\mu_{\theta ,i,j}$ and $\sigma_{\theta ,i,j}^{2}$ are the mean and variance of the *i*th representation in the *j*th dimension of the latent space. Furthermore, $\mu_{c,j}$ and $\sigma_{c,j}^{2}$ represent the mean and variance of the *c*th component of the GMM in the *j*th dimension of the latent space.

Finally, we train NNs to maximize the cost function (Equation 19) over the parameters θ,ϕ, as well as those ψ of the GMM. For the training step, we use the ADAM optimization tool [25]. The training procedure is detailed in Algorithm 1.
**Algorithm 1** VIB-GMM algorithm for unsupervised learning.1:**input:** Dataset $D:={\left\{x_{i}\right\}}_{i=1}^{N}$, parameter s≥0.2:**output:** Optimal DNN weights $\theta^{⋆},\phi^{⋆}$ and GMM parameters $\psi^{⋆}=\{\pi_{c}^{⋆}$, $\mu_{c}^{⋆}$, $\Sigma_{c}^{⋆}{\}}_{c=1}^{\left|C\right|}$.3:**initialization** Initialize θ,ϕ,ψ.4:**repeat**5:    Randomly select *b* mini-batch samples ${\left\{x_{i}\right\}}_{i=1}^{b}$ from D.6:    Draw *m* random i.i.d samples ${\left\{z_{j}\right\}}_{j=1}^{m}$ from $P_{Z}$.7:    Compute *m* samples $u_{i,j}={\tilde{f}}_{\theta }\left(x_{i},z_{j}\right)$8:    For the selected mini-batch, compute gradients of the empirical cost (Equation 20).9:    Update θ,ϕ,ψ using the estimated gradient (e.g., with SGD or ADAM).10:**until** convergence of θ,ϕ,ψ.

Once our model is trained, we assign the given dataset into the clusters. As mentioned in Section 2, we do the assignment from the latent representations, i.e., $Q_{C|U}=P_{C|X}$. Hence, the probability that the observed data $x_{i}$ belongs to the *c*th cluster is computed as follows:(24)$$p\left(c|x_{i}\right)=q\left(c|u_{i}\right)=\frac{q_{\psi^{⋆}}\left(c\right)q_{\psi^{⋆}}\left(u_{i}|c\right)}{q_{\psi^{⋆}}\left(u_{i}\right)}=\frac{\pi_{c}^{⋆}N\left(u_{i};\mu_{c}^{⋆},\Sigma_{c}^{⋆}\right)}{\sum_{c}\pi_{c}^{⋆}N\left(u_{i};\mu_{c}^{⋆},\Sigma_{c}^{*}\right)}\;,$$
where ${}^{⋆}$ indicates the optimal values of the parameters as found at the end of the training phase. Finally, the right cluster is picked based on the largest assignment probability value.

**Remark** **2.**
*It is worth mentioning that with the use of the KL approximation as given by *(Equation 22)*, our algorithm does not require the assumption $P_{C|U}=Q_{C|U}$ to hold (which is different from [19]). Furthermore, the algorithm is guaranteed to converge. However, the convergence may be to (only) local minima; and this is due to the problem *(Equation 18)* being generally non-convex. Related to this aspect, we mention that while without a proper pre-training, the accuracy of the VaDE algorithm may not be satisfactory, in our case, the above assumption is only used in the final assignment after the training phase is completed.*


**Remark** **3.**
*In [26], it is stated that optimizing the original IB problem with the assumption of independent latent representations amounts to disentangled representations. It is noteworthy that with such an assumption, the computational complexity can be reduced from $O(n_{u}^{2})$ to $O(n_{u})$. Furthermore, as argued in [26], the assumption often results only in some marginal performance loss; and for this reason, it is adopted in many machine learning applications.*


### 3.3. Effect of the Hyperparameter

As we already mentioned, the hyperparameter *s* controls the trade-off between the relevance of the representation U and its complexity. As can be seen from (Equation 19) for small values of *s*, it is the cross-entropy term that dominates, i.e., the algorithm trains the parameters so as to reproduce X as accurately as possible. For large values of *s*, however, it is most important for the NN to produce an encoded version of X whose distribution matches the prior distribution of the latent space, i.e., the term $D_{\mathrm{KL}}\left(P_{\theta }\left(U|X\right)∥Q_{\psi }\left(U\right)\right)$ is nearly zero.

In the beginning of the training process, the GMM components are randomly selected; and so, starting with a large value of the hyperparameter *s* is likely to steer the solution towards an irrelevant prior. Hence, for the tuning of the hyper-parameter *s* in practice, it is more efficient to start with a small value of *s* and gradually increase it with the number of epochs. This has the advantage of avoiding possible local minima, an aspect that is reminiscent of deterministic annealing [12], where *s* plays the role of the temperature parameter. The experiments that will be reported in the next section show that proceeding in the above-described manner for the selection of the parameter *s* helps in obtaining higher clustering accuracy and better robustness to the initialization (i.e., no need for a strong pretraining). The pseudocode for annealing is given in Algorithm 2.
**Algorithm 2** Annealing algorithm pseudocode.1:**input:** Dataset $D:={\left\{x_{i}\right\}}_{i=1}^{n}$, hyperparameter interval $\left[s_{\min},s_{\max}\right]$.2:**output:** Optimal DNN weights $\theta^{⋆},\phi^{⋆}$, GMM parameters $\psi^{⋆}=\{\pi_{c}^{⋆}$, $\mu_{c}^{⋆}$, $\Sigma_{c}^{⋆}{\}}_{c=1}^{\left|C\right|}$, assignment probability $P_{C|X}$.3:**initialization** Initialize θ,ϕ,ψ.4:**repeat**5:    Apply VIB-GMM algorithm.6:    Update ψ,θ,ϕ.7:    Update *s*, e.g., $s=\left(1+\epsilon_{s}\right)s_{\mathrm{old}}$.8:**until***s* does not exceed $s_{\max}$.

**Remark** **4.**
*As we mentioned before, a text clustering algorithm was introduced by Slonim et al. [11,12], which uses the IB method with an annealing procedure, where the parameter s is increased gradually. In [12], the critical values of s (so-called phase transitions) were observed such that if these values were missed during increasing s, the algorithm ended up with the wrong clusters. Therefore, how to choose the step size in the update of s is very important. We note that tuning s is also very critical in our algorithm, such that the step size $\epsilon_{s}$ in the update of s should be chosen carefully, otherwise phase transitions might be skipped that would cause a non-satisfactory clustering accuracy score. However, the choice of the appropriate step size (typically very small) is rather heuristic; and there exists no concrete method for choosing the right value. The choice of step size can be seen as a trade-off between the amount of computational resource spared for running the algorithm and the degree of confidence about scanning s values not to miss the phase transitions.*


## 4. Experiments

### 4.1. Description of the Datasets Used

In our empirical experiments, we applied our algorithm to the clustering of the following datasets.

MNIST: A dataset of gray-scale images of 70,000 handwritten digits of dimensions 28×28 pixel.

STL-10: A dataset of color images collected from 10 categories. Each category consisted of 1300 images of size of 96×96 (pixels) ×3 (RGB code). Hence, the original input dimension $n_{x}$ was 27,648. For this dataset, we used a pretrained convolutional NN model, i.e., ResNet-50 [27], to reduce the dimensionality of the input. This preprocessing reduced the input dimension to 2048. Then, our algorithm and other baselines were used for clustering.

REUTERS10K: A dataset that was composed of 810,000 English stories labeled with a category tree. As in [20], 4 root categories (corporate/industrial, government/social, markets, economics) were selected as labels, and all documents with multiple labels were discarded. Then, tf-idf features were computed on the 2000 most frequently occurring words. Finally, 10,000 samples were taken randomly, which were referred to as the REUTERS10K dataset.

### 4.2. Network Settings and Other Parameters

We used the following network architecture: the encoder was modeled with NNs with 3 hidden layers with dimensions $n_{x}-500-500-2000-n_{u}$, where $n_{x}$ is the input dimension and $n_{u}$ is the dimension of the latent space. The decoder consisted of NNs with dimensions $n_{u}-2000-500-500-n_{x}$. All layers were fully connected. For comparison purposes, we chose the architecture of the hidden layers, as well as the dimension of the latent space $n_{u}=10$ to coincide with those made for the DEC algorithm of [20] and the VaDE algorithm of [19]. All except the last layers of the encoder and decoder were activated with the ReLU function. For the last (i.e., latent) layer of the encoder, we used a linear activation; and for the last (i.e., output) layer of the decoder, we used the sigmoid function for MNIST and linear activation for the remaining datasets. The batch size was 100, and the variational bound (Equation 20) was maximized by the ADAM optimizer of [25]. The learning rate was initialized with 0.002 and decreased gradually every 20 epochs with a decay rate of 0.9 until it reached a small value (0.0005 in our experiments). The reconstruction loss was calculated by using the cross-entropy criterion for MNIST and the mean squared error function for the other datasets.

### 4.3. Clustering Accuracy

We evaluated the performance of our algorithm in terms of the so-called unsupervised clustering Accuracy (ACC), which is a widely used metric in the context of unsupervised learning [23]. For comparison purposes, we also present those of algorithms from the previous state-of-the-art.

For each of the aforementioned datasets, we ran our VIB-GMM algorithm for various values of the hyper-parameter *s* inside an interval $\left[s_{\min},s_{\max}\right]$, starting from the smaller valuer $s_{\min}$ and gradually increasing the value of *s* every $n_{\mathrm{epoch}}$ epochs. For the MNIST dataset, we set $\left(s_{\min},s_{\max},n_{\mathrm{epoch}}\right)=\left(1,5,500\right)$; and for the STL-10 dataset and the REUTERS10K dataset, we chose these parameters to be (1,20,500) and (1,5,100), respectively. The obtained ACC accuracy results are reported in Table 1 and Table 2. It is important to note that the reported ACC results were obtained by running each algorithm ten times. For the case in which there was no pretraining (in [19,20], the DEC and VaDE algorithms were proposed to be used with pretraining; more specifically, the DNNs were initialized with a stacked autoencoder [28]), Table 1 states the accuracies of the best case run and average case run for the MNIST and STL-10 datasets. It was seen that our algorithm outperformed significantly the DEC algorithm of [20], as well as the VaDE algorithm of [19] and GMM for both the best case run and average case run. Besides, in Table 1, the values in parentheses correspond to the standard deviations of clustering accuracies. As seen, the standard deviation of our algorithm VIB-GMM was lower than the VaDE; which could be expounded by the robustness of VIB-GMM to non-pretraining. For the case in which there was pretraining, Table 2 states the accuracies of the best case run and average case run for the MNIST and REUTERS10K datasets. A stacked autoencoder was used to pretrain the DNNs of the encoder and decoder before running algorithms (DNNs were initialized with the same weights and biases of [19]). It was seen that our algorithm outperformed significantly the DEC algorithm of [20], as well as the VaDE algorithm of [19] and GMM for both the best case run and average case run. The effect of pretraining can be observed comparing Table 1 and Table 2 for MNIST. Using a stacked autoencoder prior to running the VaDE and VIB-GMM algorithms resulted in a higher accuracy, as well as a lower standard deviation of accuracies; therefore, supporting the algorithms with a stacked autoencoder was beneficial for a more robust system. Finally, for the STL-10 dataset, Figure 4 depicts the evolution of the best case ACC with iterations (number of epochs) for the four compared algorithms.

Figure 5 shows the evolution of the reconstruction loss of our VIB-GMM algorithm for the STL-10 dataset, as a function of simultaneously varying the values of the hyperparameter *s* and the number of epochs (recall that, as per the described methodology, we started with $s=s_{\min}$, and we increased its value gradually every $n_{\mathrm{epoch}}=500$ epochs). As can be seen from the figure, the few first epochs were spent almost entirely on reducing the reconstruction loss (i.e., a fitting phase), and most of the remaining epochs were spent making the found representation more concise (i.e., smaller KL divergence). This was reminiscent of the two-phase (fitting vs. compression) that was observed for supervised learning using VIB in [29].

**Remark** **5.**
*For a fair comparison, our algorithm VIB-GMM and the VaDE of [19] were run for the same number of epochs, e.g., $n_{\mathrm{epoch}}$. In the VaDE algorithm, the cost function *(Equation 11)* was optimized for a particular value of hyperparameter s. Instead of running $n_{\mathrm{epoch}}$ epochs for s=1 as in VaDE, we ran $n_{\mathrm{epoch}}$ epochs by gradually increasing s to optimize the cost *(Equation 21)*. In other words, the computational resources were distributed over a range of s values. Therefore, the computational complexity of our algorithm and the VaDE was equivalent.*


### 4.4. Visualization on the Latent Space

In this section, we investigate the evolution of the unsupervised clustering of the STL-10 dataset on the latent space using our VIB-GMM algorithm. For this purpose, we found it convenient to visualize the latent space through application of the t-SNEalgorithm of [30] in order to generate meaningful representations in a two-dimensional space. Figure 6 shows 4000 randomly chosen latent representations before the start of the training process and respectively after 1, 5, and 500 epochs. The shown points (with a · marker in the figure) represent latent representations of data samples whose labels are identical. Colors are used to distinguish between clusters. Crosses (with an x marker in the figure) correspond to the centroids of the clusters. More specifically, Figure 6a shows the initial latent space before the training process. If the clustering is performed on the initial representations, it allows ACC as small as 10%, i.e., as bad as a random assignment. Figure 6b shows the latent space after one epoch, from which a partition of some of the points starts to be already visible. With five epochs, that partitioning is significantly sharper, and the associated clusters can be recognized easily. Observe, however, that the cluster centers seem not to have converged yet. With 500 epochs, the ACC accuracy of our algorithm reached %91.6, and the clusters and their centroids were neater, as is visible from Figure 6d.

## 5. Conclusions and Future Work

In this paper, we proposed and analyzed the performance of an unsupervised algorithm for data clustering. The algorithm used the variational information bottleneck approach and modeled the latent space as a mixture of Gaussians. It was shown to outperform state-of-the-art algorithms such as the VaDE of [19] and the DEC of [20]. We note that although it was assumed that the number of classes was known beforehand (as was the case for almost all competing algorithms in its category), that number could be found (or estimated to within a certain accuracy) through inspection of the resulting bifurcations on the associated information-plane, as was observed for the standard information bottleneck method. Finally, we mention that among the interesting research directions in this line of work, one important question pertains to the distributed learning setting, i.e., along the counterpart, to the unsupervised setting, of the recent work [31,32,33], which contained distributed IB algorithms for both discrete and vector Gaussian data models.

## Figures and Tables

**Figure 1 entropy-22-00213-f001:**
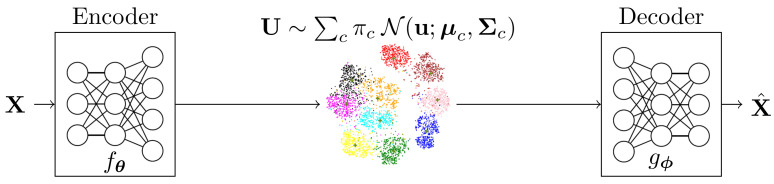
Variational Information Bottleneck with Gaussian mixtures.

**Figure 2 entropy-22-00213-f002:**
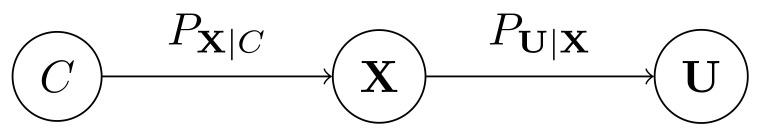
Inference network.

**Figure 3 entropy-22-00213-f003:**
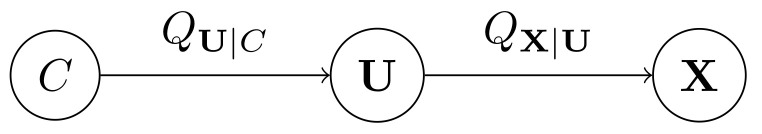
Generative network.

**Figure 4 entropy-22-00213-f004:**
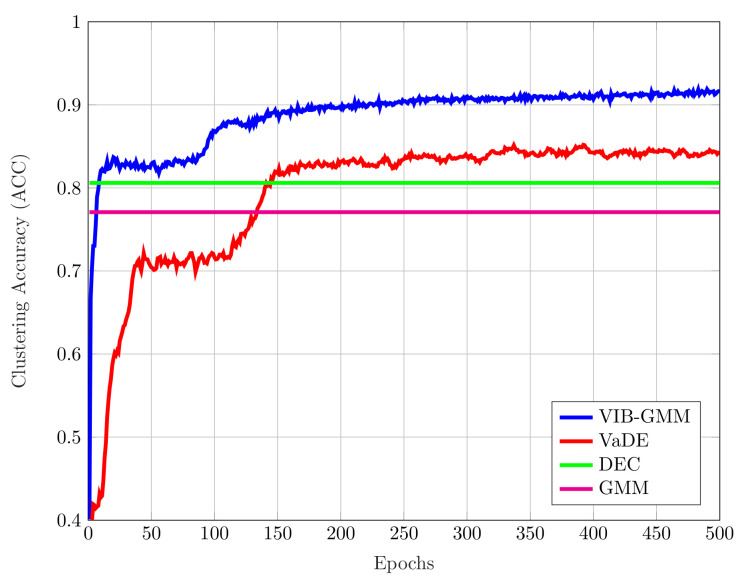
Accuracy vs. the number of epochs for the STL-10dataset.

**Figure 5 entropy-22-00213-f005:**
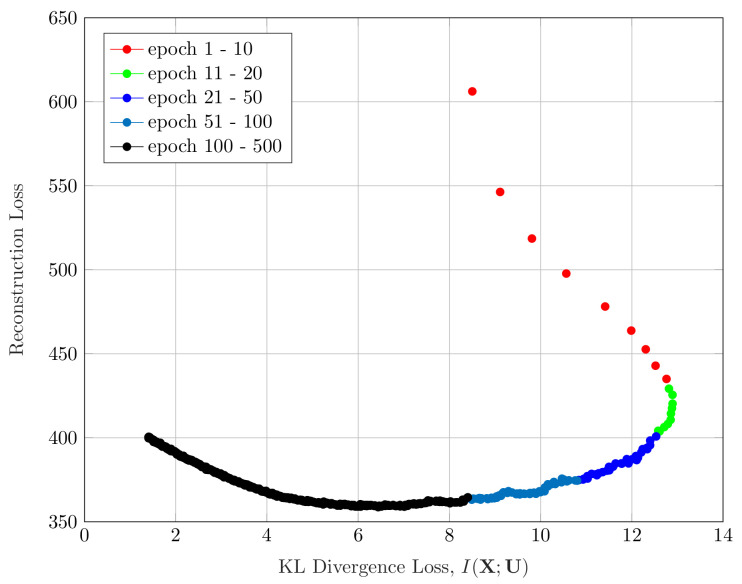
Information plane for the STL-10 dataset.

**Figure 6 entropy-22-00213-f006:**
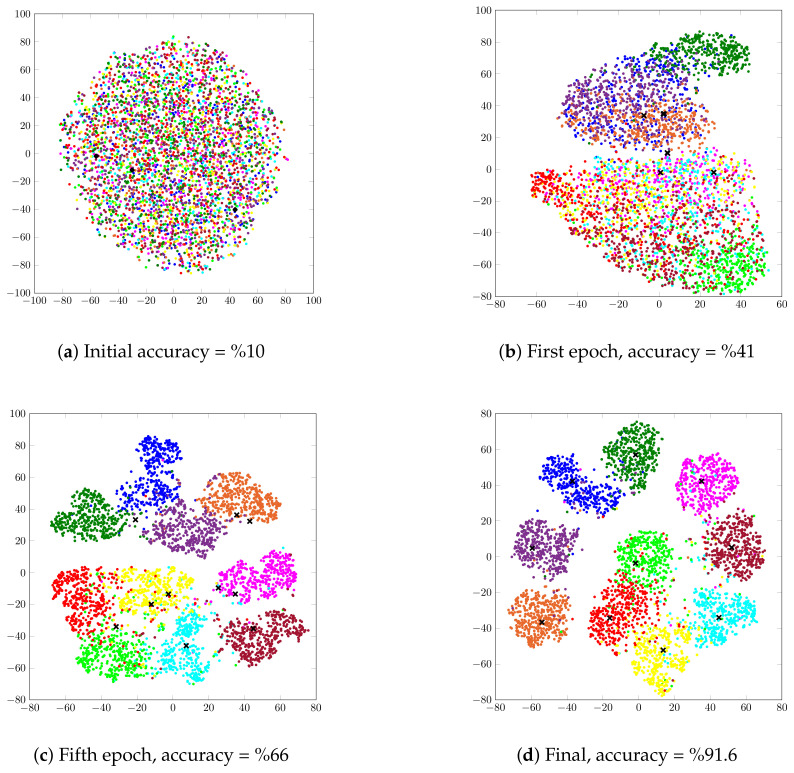
Visualization of the latent space before training; and after 1, 5, and 500 epochs.

**Table 1 entropy-22-00213-t001:** Comparison of the clustering accuracy of various algorithms. The algorithms are run without pretraining. Each algorithm is run ten times. The values in (·) correspond to the standard deviations of clustering accuracies. DEC, Deep Embedded Clustering; VaDE, Variational Deep Embedding; VIB, Variational Information Bottleneck.

	MNIST			STL-10	
	Best Run	Average Run		Best Run	Average Run
GMM	44.1	40.5 (1.5)		78.9	73.3 (5.1)
DEC				80.6 ${}^{†}$	
VaDE	91.8	78.8 (9.1)		85.3	74.1 (6.4)
**VIB-GMM**	95.1	83.5 (5.9)		93.2	82.1 (5.6)

${}^{†}$ Values are taken from VaDE [19].

**Table 2 entropy-22-00213-t002:** Comparison of the clustering accuracy of various algorithms. A stacked autoencoder is used to pretrain the DNNs of the encoder and decoder before running algorithms (DNNs are initialized with the same weights and biases of [19]). Each algorithm is run ten times. The values in (·) correspond to the standard deviations of clustering accuracies.

	MNIST			REURTERS10K	
	Best Run	Average Run		Best Run	Average Run
DEC	84.3 ${}^{‡}$			72.2 ${}^{‡}$	
VaDE	94.2	93.2 (1.5)		79.8	79.1 (0.6)
**VIB-GMM**	96.1	95.8 (0.1)		81.6	81.2 (0.4)

${}^{‡}$ Values are taken from DEC [20].

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
