# Peer review of "Variational Information Bottleneck for Unsupervised Clustering: Deep Gaussian Mixture Embedding"

_entropy, 2020, doi:10.3390/e22020213_

Round 1

Reviewer 1 Report

General remarks:
- I would prefer the problem definition of Section 2 to be part of Section 1 so that the discussion of prior art and alternatives to your proposal on page 2 is in context.
- you need to include a “discussion” or “conclusions” section.
- I have strong doubts about the evaluation technique and have requested further quantification of deviation/uncertainty, e.g. carry out 5-fold validation over the datasets and provide estimates of deviation for clustering accuracy. Similarly for training loss evolution. Do not report best-case figures in a journal paper, please.

Other:
- there are minor typos peppered around the text. Otherwise it is an excellent and very interesting read.
- the reference to PCA is not the original one, please check my remark in the ref. section.
- see my annotations throughout. I am attaching my anonymously reviewed version of the paper for your convenience.

Author Response

Thank you for your comments, please see the responses in the attachment.

Reviewer 2 Report

Although the evaluation based on publicly available benchmark data sets is an important part of the paper – and I recommend to keep this part – an evaluation with artificial data sets with known effects can provide much more insight into advantages and problems of a data analysis approach. I therefore
recommend to add an evaluation with artificial data sets. Possible aspects that could be investigated are the influence of 

(a) the dimensionality of the data set,
(b) the noise intensity,
(c) extreme outliers.

Author Response

(The authors gave the same response as above.)

Reviewer 3 Report

Please see attached review.

Author Response

(The authors gave the same response as above.)

Round 2

Reviewer 2 Report

The paper could be accepted in the current form

Author Response

For Reviewer 2, there are no issues to be addressed in this round of the revision. 

Author Response

In the pdf file, you can find the response.
